# MATHEMATICS OF NATURAL INTELLIGENCE

**Evgenii Vityaev** *
Artificial Intelligence Research Center of Novosibirsk State  University, Novosibirsk, Russia
Sobolev institute of mathematics SD RAS
 `vityaev@math.nsc.ru;`

## ABSTRACT

In the process of evolution, the brain has achieved such perfection that artificial intelligence systems do not have and which needs its own mathematics. The concept of cognitome, introduced by the academician K.V. Anokhin, as the cognitive structure of the mind – a high–order structure of the brain and a neural hypernetwork, is considered as the basis for modeling. Consciousness then is a special form of dynamics in this hypernetwork – a large-scale integration of its cognitive elements. The cognitome, in turn, consists of interconnected COGs (cognitive groups of neurons) of two types – functional systems and cellular ensembles. K.V. Anokhin sees the task of the fundamental theory of the brain and mind in describing these structures, their origin, functions and processes in them. The paper presents mathematical models of these structures based on new mathematical results, as well as models of different cognitive processes in terms of these models. In addition, it is shown that these models can be derived based on a fairly general principle of the brain works: *the brain discovers all possible causal relationships in the external world and draws all possible conclusions from them.* Based on these results, the paper presents models of: "natural" classification; theory of functional brain systems by P.K. Anokhin; prototypical theory of categorization by E. Roche; theory of causal models by Bob Rehter; theory of consciousness as integrated information by G. Tononi.

## 1   INTRODUCTION

K.V. Anokhin offers the following approach to solving the "difficult problem of consciousness" and the "brain-main problem" of neuroscience Anokhin (2021). He argues that the success of understanding the nature of consciousness critically depends on the creation of a detailed neuroscientific theory of the carrier of conscious experience – what has been called "mind" for centuries. To define the mind as a cognitive structure, K.V. Anokhin introduces the concept of "cognitome" – a high-order structure of the brain neural hypernetwork. In that case a cognition is a special form of dynamics in this hypernetwork – a large-scale integration of its cognitive elements.

By "***natural intelligence***" we will mean a cognitive hypernetwork of the brain, consisting of interconnected COGs (COgnitive Groups of neurons representing elements of subjective experience) of two types – functional systems and cellular ensembles of D. Hebb" Anokhin (2021); Vityaev (2021). K.V. Anokhin sees the task of the fundamental theory of the brain and mind in describing these systems structures, their origin, functions and processes in them Anokhin (2021); Vityaev (2021).

The purpose of this paper is to provide a mathematical description of these structures and thereby a mathematical model of the cognitome. We will first show that the basic properties and functions of these two types of COGs can be derived from a more general principle***: the brain detects all possible causal relationships in the external world and makes all possible conclusions from them***.

For this, we first analyze the concept of causality. Causality is a consequence of physical determinism: "for any isolated physical system, some of its state determines all subsequent states" Carnap (1971). In the philosophy of science, causality is reduced to prediction and explanation. "Causality

---

*This work was financial support by the State Assignment "Logical calculus and Semantics, Model theory and Computability" FWNF-2022-0011

means predictability... if all the previous situation is known, the event can be predicted ... if all the facts and laws of nature associated with this event are given" Carnap (1971). It is clear that it is not possible to know all the facts, the number of which is potentially infinite, and all the laws. In addition, humans and animals learn the laws of the external world through training (inductive inference). Therefore, causality is reduced to prediction by inductive statistical inference (section 1), when the prediction is derived from facts and statistical data with some probability.

In addition, cause-and-effect relationships in the form of statistical laws that are detected on real data or as a result of training, face the *problem of statistical ambiguity* – contradictory predictions can be derived from them (section 1) Hempel (1965; 1968). To avoid this ambiguity, Hempel introduced ***the maximum specificity requirement*** (section 1), which informally means that statistical laws should include as much available information as possible.

We solved ***the problem of statistical ambiguity*** and determined ***the most specific statistical laws*** for which it was proved (Theorem 1) that the inductive-statistical inference using them does not lead to contradictions Vityaev (2006b); Vityaev & Martinovich (2015); Vityaev & Odintsov (2019). To detect such laws, a special ***semantic probabilistic inference was developed***. In addition, we have developed ***a formal neuron model*** that satisfies the Hebb rule, which infers such laws and discovers the most specific statistical laws Vityaev (2015).

Probabilistic cause-and-effect relationships, reflecting the relationships between the properties of some object of the external world, loop on themselves, forming cellular ensembles and creating fixed points of cyclically mutually predicting attributes. These fixed points have a special meaning and reflect the ***"natural" classification*** of objects in the external world (section 2). It was noted that the "natural " classes of animals or plants differ in a potentially infinite set of properties Mill (1983). Natural scientists who built "natural" classifications noted that the construction of a "natural" classification consists in "indication" – from an infinitely large number of features, you need to move to a limited number of them, which would replace all other features Smirnov (1938). This means that in "natural" classes, features are strongly correlated, for example, if there are 128 classes and the attributes are binary, then about 7 attributes may be independent "indicator" attributes among them, since $2^7 = 128$, and other attributes can be predicted from the values of these 7 attributes. We can choose various 7-10 attributes as "indicators" and then other attributes, which are potentially infinitely many, are also predicted from these selected attributes. Therefore, there can be an exponential set of causal relationships between the attributes of objects of the "natural" classes relative to the number of attributes.

We formalize the "natural" classification by generalizing the analysis of formal concepts Ganter (2003); Ganter & Wille (2003). Formal concepts can be defined as fixed points of deterministic rules (with no exceptions) Ganter (2003). We generalize formal concepts to the probabilistic case, replacing deterministic rules with probabilistic maximally specific causal relationships and defining ***probabilistic formal concepts*** as fixed points of these maximally specific rules E.E. & Martynovich (2015); Vityaev & Martinovich (2015); Vityaev et al. (2012); Vityaev & Odintsov (2019). Due to the fact that the conclusion based on the most specific causal relationships is consistent (Theorem 1), the resulting fixed points will also be consistent and will not contain both a sign and its negation, i.e. such a definition of probabilistic formal concepts is correct.

It can be shown that probabilistic formal concepts adequately formalize the "natural" classification, and the resulting "natural" classification satisfies all the requirements that natural scientists imposed on "natural" classifications E.E. & Martynovich (2015).

In addition, this formalization (section 3) define a ***"natural" concepts*** described in the works of Eleanor Rosch Rosch (1975; 1978). The definition of "natural " concepts is based on the categorization principle of Eleanor Rosch, which postulates the structure of the perceived world: "the perceived world is not an unstructured set of equally likely properties, on the contrary, objects of the perceived world have a highly correlated structure" Rosch (1978).

On the highly correlated structure of the external world is also base the ***integrated information***, defined by G. Tononi Tononi (2010; 2012); Oizumi et al. (2014) (section 4). But G. Tononi does not have a model of the external world, and integrated information is considered as an internal property of the system of causal relationships that manifests itself in consciousness. It defines the ***concept*** notion as a system of causal relationships with maximally integrated information. Since it has no external world, it cannot be said that concept reflects "natural" concepts or "natural" objects.

Thus our formalization of "natural" classification using probabilistic formal concepts is both a formalization of "natural" concepts and integrated information. It is principally different from the formalization of the causal systems, based on Bayesian networks, as it was done in Rehder & Martin (2011), since Bayesian networks do not support cycles.

In our formalization Cognitom is a reflection of the hierarchy of "natural" classes of the external world, in which properties and objects of the class form fixed points in the form of the probabilistic formal concepts. These fixed points generates the most integrated information of G. Tononi. The algorithm of "natural" classification is based on a certain criterion of maximum consistency of causal relationships in its mutual predictions, which is close in its meaning to the integrated information.

Formalization of the second type of COGs – functional systems, is based on the consideration of purposeful behavior, which is carried out by the developing conditional (causal) relationships between actions and their results. In sections 8-11 it is shown that these conditional relationships are sufficient for modeling functional systems and developing animates.

## 2 CAUSALITY

As already mentioned, causality is reduced to prediction and explanation Carnap (1971).

There are two models for predicting some statement G:

1. Deductive-Nomological (D-N), based on facts and deductive laws.
2. Inductive-Statistical (I-S), based on facts and probabilistic laws.

**Deductive-nomological model** can be represented by the following scheme:

$$\frac{\begin{array}{c} L_1, \ldots, L_m \\ \hline C_1, \ldots, C_n \end{array}}{G}$$

1. $L_1, \ldots, L_m$ - set of laws.
2. $C_1, \ldots, C_n$ - set of facts.
3. G – predicted sentence.
4. $L_1, \ldots, L_m, C_1, \ldots, C_n \vdash G$;
5. the set $L_1, \ldots, L_m, C_1, \ldots, C_n$ is consistent.
6. $L_1, \ldots, L_m \nvdash G, C_1, \ldots, C_n \nvdash G$;
7. $L_1, \ldots, L_m$ - set of laws contain only generality quantifiers. Set $C_1, \ldots, C_n$ of facts – quantorless formulas.

**Inductive-statistical model** is similar to the previous one, with the difference that the conclusion is made with some probability, so property 7 is formulated differently:

7) the set $L_1, \ldots, L_m$ contains statistical laws. Set of facts $C_1, \ldots, C_n$– quantorless formulas.

When detecting laws based on real data*, **the problem of statistical ambiguity arises***, which consists in the fact that in the process of learning (inductive inference) we can obtain probabilistic rules from which the contradiction may be derived.

**A classic example of statistical ambiguity.**

Let's assume that there are the following statements:

(L1) – 'almost all cases of streptococcus are quickly cured by penicillin injection';

(L2) – 'Almost always penicillin-resistant streptococcal infection is not cured after penicillin injection';

(C1) – ' Jane Jones has a strep infection'.

(C2) – ' Jane Jones received penicillin injection';

(C3 – ' Jane Jones has a penicillin-resistant strep infection'.

Two contradictory statements can be deduced from this set of statements: one that explains why Jane Jones will recover quickly (E), and the other that explains why Jane Jones will not recover quickly (E).

| Explanation 1 | | Explanation 2 | |
|---|---|---|---|
| L1 | | L2 | |
| C1, C2 | [r] | C2, C3 | [r] |
| E | | $\neg E$ | |

The conditions of both explanations do not contradict each other, both of them can be simultaneously true. However, their conclusions contradict each other.

To avoid contradictions, Hempel Hempel (1968) introduced the ***requirement of maximum specificity*** for statistical laws:

The rule F $\Rightarrow$ G is ***maximally specific*** in the state of knowledge K,

| F $\Rightarrow$ G, p(G;F)= r | |
|---|---|
| F($a$) | [r] |
| G($a$) | |

if for every class H for which both the following statements $\forall x(H(x) \Rightarrow F(x))$, H($a$) , belong to K, there exists a statistical law p(G; H) = r' in K such that r = r'. The idea behind the maximum specificity requirement is that if F and H both contain object $a$, and H is a subset of F, then H has more specific information about object $a$, and, therefore, the law p(G;H) should be preferred to the law p(G;F), but the law p(G;H) must have the same probability as the law p(G;F). So any specification of the set of objects H can not change the probability r.

Despite the fact that Hempel gave a fairly precise informal definition of the maximum specificity requirement, neither he nor his followers proved that I-S inference, based on the most specific rules, is consistent.

We solved the problem of statistical ambiguity and defined the most specific statistical laws for which we prove (Theorem 1) that an inductive-statistical inference, based on such laws does not lead to contradictions Vityaev (2006b); E.E. & Martynovich (2015); Vityaev & Martinovich (2015); Vityaev & Odintsov (2019).

## 3   WHAT IS A "NATURAL" CLASSIFICATION?

The highly correlated structure of the external world was revealed in two theories – "natural" classification and "natural" concepts. At the same time, "natural" classification refers to the structure of objects in the external world, and "natural" concepts, studied in cognitive science, refer to the perception of these "natural" objects.

The first sufficiently detailed analysis of the "natural" classification belongs to J. S. Mill Mill (1983). First, let's separate "artificial" classifications from "natural" ones by J. S. Mill:

1. artificial classifications - *"Let's take any attribute, and if some things have it, and others do not, then we can base the division of all things into two classes on it"*.

2. natural classifications - *"But if we turn to a class "animal" or "plant", and if we look at what features the individuals embraced by a given class differ from those not included in it, we will find that in this respect some classes differ greatly from others. ... have such a large number of characteristics that they cannot be enumerated"* .

J. S. Mill defines the "natural" classification as follows: *"Natural groups ... are defined by features, ... taking into account not only the features that are absolutely common to all the objects included in the group, but the whole set of those features, of which all are found in the majority of these objects, and the majority – in all. Consequently, our concept of a class – the image that this class is represented in our mind - is the concept of a certain pattern that has the majority of the characteristics of this class."*

The problem of defining of "natural" classifications is still discussed in the literature Nat (2013). However, from our point of view, there is no sufficiently adequate formalization of the "natural" classification.

## 4   WHAT IS A "NATURAL" CONCEPTS?

In the introduction it was mentioned a categorization principle of Eleanor Rosch. Thus, basic objects are information-rich bundles of observed properties, that create categorization: "Categories can be viewed in terms of their clear cases if the perceiver places emphasis on the *correlational structure of perceived attributes* ... By prototypes of categories we have generally meant the clearest cases of category membership" Rosch (1975; 1978). Later, Eleanor Rosch's theory of "natural" concepts was called prototype theory.

In further research, it was found that models based on traits, similarities, and prototypes are not sufficient to describe "natural" concepts. It is necessary to take into account theoretical, causal, and ontological knowledge related to concept objects. For example, people not only know that birds have wings, can fly and build nests in trees, but also that birds build nests in trees because they can fly, and fly because they have wings.

Considering these studies, Bob Rehder put forward the causal-model theory, according to which: "people's intuitive theories about categories of objects consist of a model of the category in which both a category's features and the causal mechanisms among those features are explicitly represented" Rehder (2003). In the theory of causal models, the relation of an object to a category is no longer based on the set of features and proximity by features, but on the similarity of the generating causal mechanism: "Specifically, a to-be-classified object is considered a category member to the extent that its features were likely to have been generated by the category's causal laws, such that combinations of features that are likely to be produced by a category's causal mechanisms are viewed as good category members and those unlikely to be produced by those mechanisms are viewed as poor category members" Rehder (2003).

Bob Rehder used Bayesian networks to represent causal models Rehder & Martin (2011). However, they do not support cycles and therefore cannot model cyclic causal relationships. Our proposed formalization by the probabilistic formal concepts directly models cyclic causal relationships represented by fixed points of predictions based on causal relationships Vityaev (2006b); Vityaev et al. (2012); Vityaev & Martinovich (2015); E.E. & Martynovich (2015).

## 5   INTEGRATED INFORMATION THEORY BY G. TONONI

G. Tononi's theory of integrated information is also based on the highly correlated structure of the external world G.Tononi Tononi (2010; 2012); Oizumi et al. (2014). Integrated information considered by G. Tononi as a property of a system of cyclic causal relationships: «Indeed, a "snapshot" of the environment conveys little information unless it is interpreted in the context of a system whose complex causal structure, over a long history, has captured some of the causal structure of the world, i.e. long-range correlations in space and time» Tononi (2012).

G. Tononi describes the relationship of integrated information with reality as follows: "Cause-effect matching ... measures how well the integrated conceptual structure ... fits or 'matches' the cause-effect structure of its environment", "... matching should increase when a system adapts to an environment having a rich, integrated causal structure. Moreover, an increase in matching will tend to be associated with an increase in information integration and thus with an increase in consciousness» Tononi (2010; 2012).

G. Tononi defines consciousness as a primary concept that has the following phenomenological properties: composition, information, integration, exclusion Tononi (2010; 2012); Oizumi et al. (2014). We present the formulations of these properties together with our interpretation of these properties (given in parentheses) from the point of view of the "natural" classification.

1. composition – elementary mechanisms (causal interactions) can be combined into higher-order ones ("natural" object classes form a hierarchy).

2. information – only mechanisms that specify 'differences that make a difference' within a system ≪count≫ (only the system of "resonating" causal relationships forming the class is significant).

3. integration – only information irreducible to non-interdependent components counts (only a system of "resonant" causal relationships is significant, which is not reduced to information of individual components, indicating a perception of a highly correlated structure of a "natural" object).

4. exclusion – only maxima of integrated information count (only values of features that are maximally interconnected by causal relationships form an "image" or "prototype").

Unlike G. Tononi, we consider these properties not as internal properties of the system, but as the ability of the system to reflect the "natural" classification of objects of the external world, and consciousness – as the ability of a complex hierarchical reflection of the "natural" classification of the external world.

## 6   PROBABILISTIC FORMAL CONCEPTS

We present our formalization of the basic concepts: Cartwright definitions of probabilistic causality, semantic probabilistic inference, maximally specific causality, and probabilistic formal concepts. This section follows the works Vityaev (2006b); Vityaev & Martinovich (2015); Ganter (2003); Ganter & Wille (2003).

**Definition 1**. *A formal context* $K = (G, M, I)$ is a triple, where $G$ and $M-$ arbitrary set of objects and attributes, and $I \subseteq G \times M-$ is a binary relation that expresses the belonging of an attribute to an object.

In a formal context, the derived operators play a key role – they link subsets of objects and attributes of the context.

**Definition 2**. $A \subseteq G, B \subseteq M$, then:

1. $A^{\uparrow} = \{m \in M \mid \forall g \in A, (g, m) \in I\}$
2. $B^{\downarrow} = \{g \in G \mid \forall m \in B, (g, m) \in I\}$

**Definition 3.** A pair $(A, B)$ is a formal concept if $A^{\uparrow} = B$ and $B^{\downarrow} = A$.

In the framework of logic, we will consider only finite contexts.

**Definition 4.** For a context $K = (G, M, I)$, we define $\Omega_K$ a context signature, that contains predicate symbols for each $m \in M$, $K \models m(x) \Leftrightarrow (x, m) \in I$.

**Definition 5.** For the signature $\Omega_K$, we define the following variant of first-order logic:

1. $X_K$ - set of variables;
2. $At_K-$ set of atomic formulas (atoms) $m(x)$, $m \in \Omega_K$, $x \in X_K$;
3. $L_K$ – set of literals that includes all atoms $m(t)$ and their negations $\neg m(t)$;
4. $\Phi_K$ – set of formulas defined inductively: a literal is a formula, and for any $\Phi, \Psi \in \Phi_K$ expressions $\Phi \wedge \Psi$, $\Phi \vee \Psi$, $\Phi \rightarrow \Psi$, $\neg \Phi$ is also a formula.

We define conjunction $\wedge L$ and negation $\neg L = \{\neg P \mid P \in L\}$ of a set of literals $L \subseteq L_K$.

**Definition 6.** A single signature $\Omega_K$ element $\{g\}$ forms the model $K_g$ of this object. The truth of the formula $\phi$ on the model $K_g$ is defined as $g \models \phi \Leftrightarrow K_g \models \phi$.

**Denition 7.** We define *a probability measure* $\mu$ on a set G in the Kolmogorov sense. Then we can define the probability measure on the set of formulas as:

$$\nu : \Phi_K \rightarrow [0, 1], \ \nu(\phi) = \mu(\{g \mid g \models \phi\}).$$

We assume that there are no non-essential objects in the context, such as $\mu(\{g\}) = 0, \ g \in G$.

**Definition 8.** Let $\{H_1, H_2, \ldots, H_k, C\} \in L_K$, $C \notin \{H_1, H_2, \ldots H_k\}, k \geq 0$.

1. *There is a relationship $R = (H_1 \wedge H_2 \wedge \ldots \wedge H_k \rightarrow C)$;*
2. *The premise $R^{\leftarrow}$ of the relation R is a set of literals $\{H_1, H_2,..., H_k\}$;*
3. *The conclusion of the relationship is $R^{\rightarrow} = C$;*
4. *The length of the relationship is $|R^{\leftarrow}|$.*

**Definition 9.** *The probability $\eta$ of the relation R – is a quantity*

$$\eta(R) = \nu(R^{\rightarrow}|R^{\leftarrow}) = {\nu(R^{\leftarrow} \wedge R^{\rightarrow})}/{\nu(R^{\leftarrow})}.$$

If the denominator $\nu(R^{\leftarrow})$ of the ratio is 0, then the probability is undefined.

**Definition 10.** The relation $R_1$ is a sub-relation of the relation $R_2$, denoted as $R_1 \sqsubset R_2$, if $R_1^{\rightarrow} = R_2^{\rightarrow}$, $R_1^{\leftarrow} \subset R_2^{\leftarrow}$.

**Definition 11.** The relation $R_1$ *refines* the relation $R_2$, denoted as $R_2 < R_1$, if $R_2 \sqsubset R_1$ and $\eta(R_1) > \eta(R_2)$.

**Definition 12.** The relation R is *a probabilistic causal relationship*, if each $\tilde{R}$ is satisfied $(\tilde{R} \sqsubset R) \Rightarrow (\tilde{R} < R)$.

The probabilistic causality given by Cartwright [20] with respect to a certain background can be formulated in the above terms as follows. If the premise $R^{\leftarrow}$ of the relation R is a set of literals $\{H_1, H_2,..., H_k\}$ and we consider this set as abackground, then each literal of the premise is a probabilistic reason for the conclusion $R^{\rightarrow}$ the with respect to this background, that is

$\nu(R^{\rightarrow}/R^{\leftarrow}) > \nu(R^{\rightarrow}/(R^{\leftarrow} \backslash H))$ for everyone $H \in \{H_1, H_2,..., H_k\}$.

It is easy to see that this definition follows from definition 12.

**Definition 13.** *The strongest probabilistic causal relationship* is a relation R for which there is no such probabilistic causal relationship $\tilde{R}$ that $(\tilde{R} > R)$.

**Definition 14.** *The Semantic Probabilistic Inference* (SPI) of predictions of a certain literal C is a sequence of probabilistic causal relationships $R_0 < R_1 < R_2... < R_m$, $R_0^{\rightarrow} = R_1^{\rightarrow} = R_2^{\rightarrow}... = R_m^{\rightarrow} = C$, $R_0^{\leftarrow} = \emptyset$, $R_m$ – strongest probabilistic causal relationship.

**Definition 15.** The tree of *semantic probabilistic inference* Tree(C) of a certain literal C – is a collection of all SPI predictions of the literal C.

**Definition 16.** *The most specific causal relation* for predicting a certain C – is the strongest probabilistic causal relation of the Tree(C) with the maximum conditional probability.

Let us denote by MSCR the set of all maximally specific causal relations for predicting some literal C. By *the system of causal relations* we mean any subset $\mathcal{R} \subseteq$ MSCR.

**Definition 17.** Define *the prediction operator* for the system $\mathcal{R}$ as:

$$\Pi_{\mathcal{R}}(L) = L \cup \{C \mid \exists R \in \mathcal{R} : R^{\leftarrow} \subseteq L, R^{\rightarrow} = C\}.$$

**Definition 18.** By the *closure* of a set of literals L we mean the smallest fixed point of the prediction operator containing L:

$$\Pi_{\mathcal{R}}^{\infty}(L) = \bigcup_{k \in \mathbb{N}} \Pi_{\mathcal{R}}^{k}(L).$$

A set of literals L *is consistent* if it does not contain both the atom C and its negation $\neg C$. A set of literals L is *compatible* if $\nu(\wedge L) \neq 0$.

**Theorem 1.** If L is compatible, then $\Pi_{\mathcal{R}}(L)$ compatible and consistent for any system $\mathcal{R}$.

**Proof of Theorem 1**. Below are the three lemmas necessary to prove Theorem 1, and then the proof of the theorem itself.

**Lemma 1**. Any probabilistic causal relationship $C = (A_1 \& ... \& A_k \Rightarrow A_0)$ belongs to some semantic probabilistic output of the letters $A_0$, and, consequently, to the tree of semantic probabilistic inference of the letter $A_0$.

**Proof.** For probabilistic causal relationship C $= (A_1\&...\&A_k \Rightarrow A_0)$, $k \geq 1$we find a sub-relationship that is a probabilistic causal relationship. This always exists, because the rule C $= (\Rightarrow A_0)$ is a probabilistic causal relationship. The conditional probability of this sub-relationship will be strictly less than the conditional probability of the causation itself. We add it as a previous rule for semantic probabilistic inference and continue the procedure.

**Lemma 2**. For any rule C $= (A_1\&...\&A_k \Rightarrow A_0)$, $k \geq 0$,$A_0 \notin \{A_1\&...\&A_k\}$, $\eta(A_1\&...\&A_k) > 0$ there is a probabilistic causal relationship $C' = (B_1\&...\&B_{k'} \Rightarrow A_0)$, $B_1\&...\&B_{k'} \subseteq A_1\&...\&A_k$, $k' \leq k$, for which $\mu(C') \geq \mu(C)$.

**Proof.** A rule C $= (A_1\&...\&A_k \Rightarrow A_0)$ is either a probabilistic causal relationship and then it is the desired one, or, by virtue of the definition of a probabilistic causal relationship, there is sub-rule $(\tilde{R} \sqsubset R)$, $\tilde{R} = (P_1\& \ldots \&P_{k'} \Rightarrow A_0)$, $k' \geq 0$, $\{P_1, \ldots, P_{k'}\} \subset \{A_1, \ldots, A_k\}$, $k' < k$ for which $\mu(\tilde{R}) \geq \mu(A)$. Similarly for a rule $R'$, either it is a probabilistic causal relationship, or there is a sub-rule with similar properties for it, etc. Since the rule C $= (\Rightarrow A_0)$is a probabilistic causal relationship, the process will stop.

**Lemma 3**. If the inequality $\eta(G/\bar{A}\&\neg\bar{B}) > \eta(G/\bar{A})$ is true for the rules A $= (\bar{A} \Rightarrow G)$, B $= (\bar{B} \Rightarrow \neg G)$, $\bar{A} = A_1\&...\&A_k$, $\bar{B} = B_1\&...\&B_m$, $\eta(\bar{A}\&\neg\bar{B}) > 0$, $k \geq 0$, $m > 0$, then there exists a rule with a strictly higher conditional probability than the rule A.

**Proof.** Let's rewrite the conditional probability

$$\eta(G/\bar{A}\&\neg\bar{B}) = \eta(G/\bar{A}\&(\neg B_1 \vee ... \vee \neg B_m)).$$

Let us represent a disjunction $\neg B_1 \vee ... \vee \neg B_m$ as a disjunction of conjunctions $\overset{i=(1,...1,0)}{\underset{i=(0,...,0)}{\vee}}(B_1^{i_1}\&...\&B_m^{i_m})$, $i = (i_1,...,i_m)$, $i_1,...,i_m \in \{0,1\}$, zero means the presence of negation in the corresponding atom, and one – means the absence of negation. The disjunction does not include the tuple (1,...,1) corresponding to the conjunction $B_1\&...\&B_m$.

Then the conditional probability $\eta(G/\bar{A}\&(\neg B_1 \vee ... \vee \neg B_m))$ is rewritten as $\eta\left(G/\overset{i=(1,...1,0)}{\underset{i=(0,...,0)}{\vee}}(\bar{A}\&B_1^{i_1}\&...\&B_m^{i_m})\right)$.

We prove that if $\eta(G/\bar{A}\&\neg\bar{B}) > \eta(G/\bar{A})$, then one of the inequalities also holds

$$\eta(G/\bar{A}\&B_1^{i_1}\&...\&B_m^{i_m}) > \eta(G/\bar{A}), (i_1,...,i_m) \neq (1,...,1).$$

To the contrary, assume that all inequalities are met simultaneously

$$\eta(G/\bar{A}\&B_1^{i_1}\&...\&B_m^{i_m}) \leq \eta(G/\bar{A}), (i_1,...,i_m) \neq (1,...,1)$$

in cases where $\eta(\bar{A}\&B_1^{i_1}\&...\&B_m^{i_m}) > 0$.

Since $\eta(\bar{A}\&\neg\bar{B}) > 0$, there are cases when $\eta(\bar{A}\&B_1^{i_1}\&...\&B_m^{i_m}) > 0$.

Then

$$\eta(G\&\bar{A}\&B_1^{i_1}\&...\&B_m^{i_m}) \leq \eta(G/\bar{A})\eta(\bar{A}\&B_1^{i_1}\&...\&B_m^{i_m}), (i_1,...,i_m) \neq (1,...,1),$$

$$\eta\left(G/\overset{i=(1,...1,0)}{\underset{i=(0,...,0)}{\vee}}(\bar{A}\&B_1^{i_1}\&...\&B_m^{i_m})\right) = \frac{\eta\left(\overset{i=(1,...1,0)}{\underset{i=(0,...,0)}{\vee}}(G\&\bar{A}\&B_1^{i_1}\&...\&B_m^{i_m})\right)}{\eta\left(\overset{i=(1,...1,0)}{\underset{i=(0,...,0)}{\vee}}(\bar{A}\&B_1^{i_1}\&...\&B_m^{i_m})\right)} =$$

$$\frac{\sum_{i=(0,...,0)}^{i=(1,...1,0)}\eta(G\&\bar{A}\&B_1^{i_1}\&...\&B_m^{i_m})}{\sum_{i=(0,...,0)}^{i=(1,...1,0)}\eta(\bar{A}\&B_1^{i_1}\&...\&B_m^{i_m})} \leq \frac{\eta(G/\bar{A})\sum_{i=(0,...,0)}^{i=(1,...1,0)}\eta(\bar{A}\&B_1^{i_1}\&...\&B_m^{i_m})}{\sum_{i=(0,...,0)}^{i=(1,...1,0)}\eta(\bar{A}\&B_1^{i_1}\&...\&B_m^{i_m})} = \eta(G/\bar{A}),$$

which contradicts the inequality $\eta(G/\bar{A}\&\neg\bar{B}) > \eta(G/\bar{A})$. Therefore, our assumption is not true and there is a rule of the form

$$\bar{A}\&B_1^{i_1}\&...\&B_m^{i_m} \Rightarrow G, (i_1,...,i_m) \neq (1,...,1)$$

having a strictly higher conditional probability than A .

**Proof of the theorem**.

First we prove that each time a rule from $\mathrm{P} \subset \mathrm{MSCR}$ is applied, we again get a compatible set of letters. Suppose, on the contrary, that applying of a certain rule $\mathrm{A} = (\mathrm{A}_1 \& ... \& A_k \Rightarrow \mathrm{G})$, $\{\mathrm{A}_1, ..., A_k\} \subset L$, $k > 1$ to a set of letters $L = \{L_1, ..., L_n\}$ outputs the letter G, for which $\nu(\mathrm{L}_1 \& ... \& L_n \& G) = 0$.

Since for rules MS!R the inequalities $\eta(\mathrm{G}/\mathrm{A}_1 \& ... \& A_k) > \nu(\mathrm{G})$, $\nu(\mathrm{A}_1 \& ... \& A_k) > 0$, $\nu(\mathrm{G}) > 0$ are satisfied, then

$$\nu(\mathrm{G} \& \mathrm{A}_1 \& ... \& A_k) > \nu(\mathrm{G})\nu(\mathrm{A}_1 \& ... \& A_k) > 0.$$

Add negations of letters $\{B_1, ..., B_t\} = \{\mathrm{L}_1 \& ... \& L_n\} \backslash \{\mathrm{A}_1, ..., A_k\}$ to the rule A, then we get the rule $(\mathrm{A}_1 \& ... \& A_k \& \neg(B_1 \& ... \& B_t) \Rightarrow \mathrm{G})$.

Let's denote $\bar{\mathrm{A}} = \mathrm{A}_1 \& ... \& A_k$, $\bar{B} = B_1 \& ... \& B_t$, $\bar{\mathrm{L}} = \mathrm{L}_1 \& ... \& L_n$.

By the assumption $\nu(\mathrm{L}_1 \& ... \& L_n \& G) = 0$ of and $\nu(\bar{\mathrm{A}} \& \bar{B}) = \nu(\bar{\mathrm{L}}) > 0$. Let us prove that in this case $\nu(\bar{\mathrm{A}} \& \neg\bar{B}) > 0$. Suppose the opposite that $\nu(\bar{\mathrm{A}} \& \neg\bar{B}) = 0$, then $\nu(\mathrm{G} \& \bar{\mathrm{A}} \& \neg\bar{B}) \leq \nu(\bar{\mathrm{A}} \& \neg\bar{B}) = 0$, where it follow that

$$0 = \nu(\bar{\mathrm{L}} \& G) = \nu(\mathrm{G} \& \bar{\mathrm{A}} \& \bar{B}) = \nu(\mathrm{G} \& \bar{\mathrm{A}}) - \nu(\mathrm{G} \& \bar{\mathrm{A}} \& \neg\bar{B}) = \nu(\mathrm{G} \& \bar{\mathrm{A}}) > 0.$$

We got a contradiction. Then

$$\nu(\mathrm{G}/\bar{\mathrm{A}} \& \neg\bar{B}) = \frac{\nu(\mathrm{G} \& \bar{\mathrm{A}} \& \neg\bar{B})}{\nu(\bar{\mathrm{A}} \& \neg\bar{B})} = \frac{\nu(\mathrm{G} \& \bar{\mathrm{A}}) - \nu(\mathrm{G} \& \bar{\mathrm{A}} \& \bar{B})}{\nu(\bar{\mathrm{A}}) - \nu(\bar{\mathrm{A}} \& \bar{B})} =$$
$$\frac{\nu(\mathrm{G} \& \bar{\mathrm{A}}) - \nu(\mathrm{G} \& \bar{\mathrm{L}})}{\nu(\bar{\mathrm{A}}) - \nu(\bar{\mathrm{A}} \& \bar{B})} = \frac{\nu(\mathrm{G} \& \bar{\mathrm{A}})}{\nu(\bar{\mathrm{A}}) - \nu(\bar{\mathrm{A}} \& \bar{B})} > \frac{\nu(\mathrm{G} \& \bar{\mathrm{A}})}{\nu(\bar{\mathrm{A}})} = \nu(\mathrm{G}/\mathrm{A}_1 \& ... \& A_k).$$

Then, by lemmas 2, 4, 5, we get that there is a probabilistic causal relationship with a higher conditional probability than the rule A, which contradicts the maximum specificity of the rule A.

Since the set of letters $\mathrm{L}_1, ..., L_n, G$ is compatible and $\nu(\mathrm{L}_1 \& ... \& L_n \& G) > 0$, then it is consistent, since if it contained both letters G and $\neg G$, then its probability would be zero.

**Definition 19.** *A probabilistic formal concept* on the context K – is a pair (A, B) satisfying the following conditions:

$$\Pi_{\mathcal{R}}^{\infty}(B) = B, \ A = \underset{\Pi_{\mathcal{R}}^{\infty}(C)=B}{\cup} C^{\downarrow}.$$

The definition of a set A is based on the following theorem linking probabilistic and standard formal concepts on the context K.

**Theorem 2.** Let $\mathrm{K} = (\mathrm{G}, \mathrm{M}, \mathrm{I})$ be a formal context, then:

1. If (A,B) is a formal concept on K, then there exists a probabilistic formal concept (S,T) on K such that $A \subseteq S$, .

2. If (S,T) is a probabilistic formal concept on K, then there exists a family $\mathcal{C}$ of formal concepts on K such that

$$\forall(A, B) \in \mathcal{C} \ (\Pi_{\mathcal{R}}^{\infty}(B) = T), \ S = \underset{(A,B)\in\mathcal{C}}{\cup} A.$$

**Proof of the theorem**:

1. Put $T = \Pi_{\mathcal{R}}^{\infty}(B)$ and $S = \underset{\Pi_{\mathcal{R}}^{\infty}(C)=B}{\cup} C^{\downarrow}$. Then it is obvious that $B \subseteq T$ and $A \subseteq S$.

2. Define $P = \{Q | \Pi_{\mathcal{R}}^{\infty}(Q) = T, Q \subseteq M\}$. By $P$, we construct a set of strict concepts $\mathcal{C} = \{(Q^{\downarrow}, Q^{\downarrow\uparrow}) | Q \in P\}$. Then, if we define $S = \bigcup_{(A,B)\in\mathcal{C}} A$, then the conditions of the theorem are satisfied.

## 7 CAUSALITY AND THE FORMAL NEURON MODEL

We present a formal model of a neuron that discover maximally specific conditional connections.

By *the information* received at the brain's "input" we will understand all the stimulation perceived by the brain: motivational, situational, trigger, sanctioning, reverse afferentation about actions performed, arriving at the "input" via collaterals, and so on. From the ecological theory of perception by J. Gibson Gibson (1988), it follows that information can be understood as any characteristic of the energy flow of light, sound, etc., coming to the "input" of the brain.

We define the information transmitted by the excitation of a certain nerve fiber to the synapses of a neuron by single predicates

$$P_j^i(a) \Leftrightarrow (x_i(a) = x_{ij}), i = 1, ..., n; \; j = 1, ..., n_i,$$

where $x_i(a)$- information, and $x_{ij}$- its value in the current situation (on the object) $\boldsymbol{a}$. If information is transmitted to the excitatory synapse, then it is perceived by the neuron as the truth of the predicate $P_j^i(a)$, if to the inhibitory synapse, then as the falsity $\neg P_j^i(a)$ of the predicate. We define the excitation of a neuron in situation $\boldsymbol{a}$ and the transmission of this excitation to the axon of the neuron by a predicate $P_0(a)$. If the neuron is inhibited in situation $\boldsymbol{a}$, then we define this situation as predicting the negation of the predicate $\neg P_0(a)$. Predicates $P_j^i(a)$, $P_0(a)$ and their negations $\neg P_j^i(a)$, $\neg P_0(a)$ are literals (atomic statements or their negations), which we denote as $a, b, c, ... \in L$, where $L$ is the set of all literals in the dictionary $\{P_0\} \cup \{P_j^i\}$, $i = 1, ..., n; \; j = 1, ..., n_i$.

Each neuron has its own receptive field, which excites it unconditionally. The initial (before any training) semantics of the predicate $P_0$ is this receptive field.

We assume that the formation of conditional connections at the neuron level occurs according to the Hebb rule Hebb (1949). We formalize the Hebb rule using semantic probabilistic inference (definition 14), which is fundamentally different from other formalizations in that it reveals maximally specific causal relationships.

Therefore, a neuron in the process of semantic probabilistic inference (see Figure 1) detects a set of $\{R\}$ maximally specific causal relationships of the form:

$$R = (a_1 \& ... \& a_k \Rightarrow b), a_1, ..., a_k, b \in L,$$

where $a_1, ..., a_k$ are the predicates coming to the synapses of the dendrites of the neuron, and $b$ is the predicate $P_0(a)$ or $\neg P_0(a)$ axon of the neuron. The tree of semantic probabilistic inference in Fig. 1 and the structure of the neuron, presented in Fig. 2, are similar, so it is easy to imagine that neuron implements such inference.

Let's describe the properties of the resulting formal neuron model:

1. the neuron performs "closure of conditional connections". When conditional stimuli are detected that allow predicting with some probability the excitation/inhibition of a neuron, conditional connection is formed that forms the corresponding rule. When new stimuli are detected that allow predicting the excitation/inhibition of a neuron with even greater probability, it attaches them to this conditional connection.

2. the neuron is activated or inhibited according to the most probable rules. This is confirmed by the fact that in the process of discovering conditional connections, as well as when closing conditional connections at the neuron level, the speed of the neuron's response to a conditional signal is higher, the higher the probability of a conditional connection. Since the most specific rules that take into account all available information are at the same time the most probable, the prediction (neuron excitation) is carried out according to them;

3. the prediction made by neuron according to the most specific rules is consistent. Therefore, the neuron learns to predict without contradictions – either its excitatory, or inhibitory maximally specific rules are triggered, but not simultaneously.

4. Figure 2 shows several semantic probabilistic inferences made by a neuron. For example, a conditional connection $(b \Leftarrow a_1^1 \& a_2^1)$ is enhanced by new stimuli $a_3^1 \& a_4^1$ providing the connection $(b \Leftarrow a_1^1 \& a_2^1 \& a_3^1 \& a_4^1)$, if the stimuli $a_3^1 \& a_4^1$ increase the conditional probability of predicting the firing of a neuron $b$.

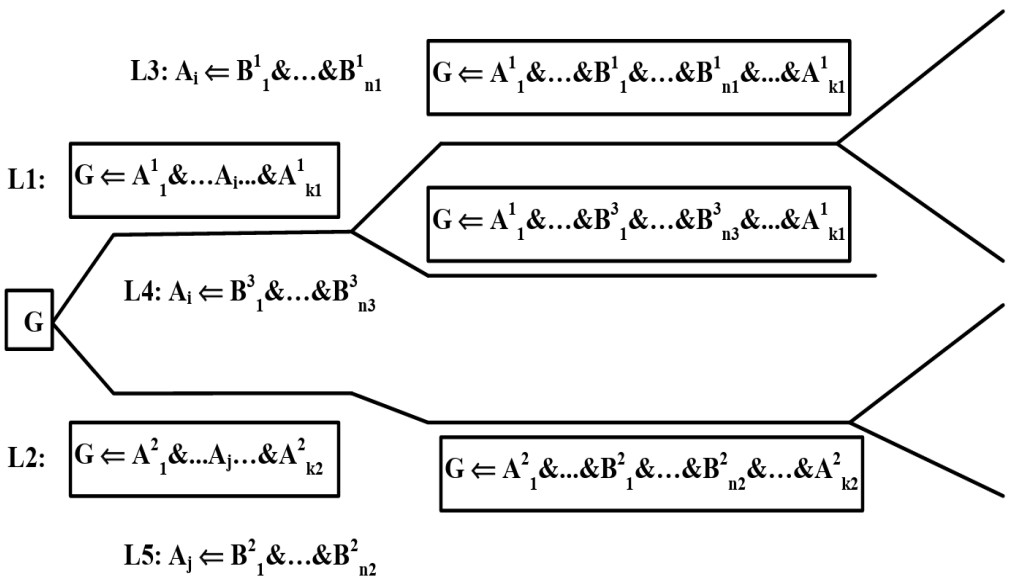

Figure 1: Semantic probabilistic inference tree

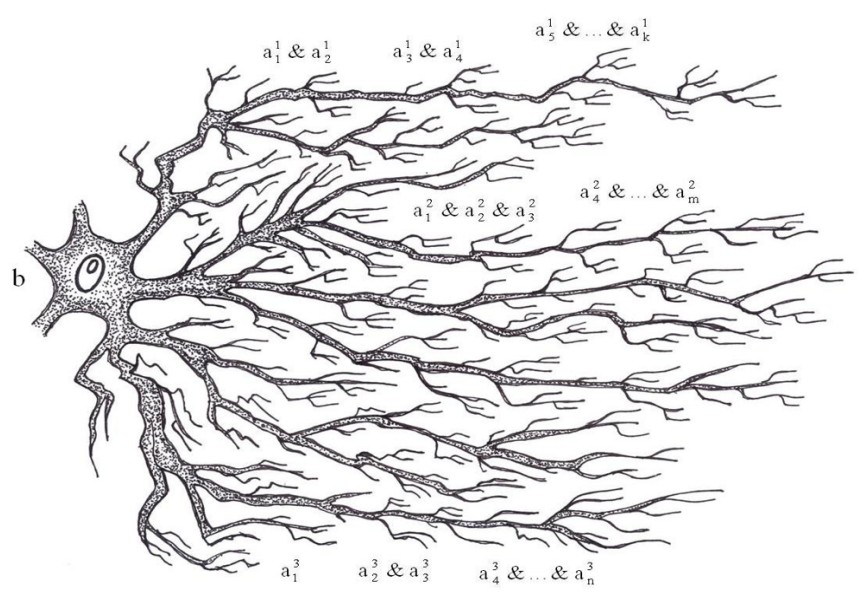

Figure 2: Formal model of neuron

$(b \Leftarrow a_1^2 \& a_2^2 \& a_3^2) \mid (b \Leftarrow a_1^2 \& a_2^2 \& a_3^2 \& a_4^2 \& ... \& a_m^2); (b \Leftarrow a_1^2 \& a_2^2 \& a_3^2) \mid (b \Leftarrow a_1^2 \& a_2^2 \& a_3^2 \& a_4^2 \& ... \& a_m^2); a)(b \Leftarrow a_1^1 \& a_2^1 \& a_3^1 \& a_4^1 \& a_5^1 \& ... \& a_k^1);$

$(b \Leftarrow a_1^3) \mid (b \Leftarrow a_1^3 \& a_2^3 \& a_3^3) \mid (b \Leftarrow a_1^3 \& a_2^3 \& a_3^3 \& a_4^3 \& ... \& a_n^3). (b \Leftarrow a_1^3) \mid (b \Leftarrow a_1^3 \& a_2^3 \& a_3^3) \mid (b \Leftarrow a_1^3 \& a_2^3 \& a_3^3 \& a_4^3 \& ... \& a_n^3). b)$

c)

Semantic probabilistic inference and the Discovery software system that implements it as a neuron model, have been successfully used to solve a number of applied problems Kovalerchuk & Vityaev (2000); Vityaev (2006a); Vityaev & Kovalerchuk (2017).

## 8    TASK CONCEPT OF AND THE FOUNDATIONS OF MATHEMATICS

We present an analysis of the ***desire*** concept given in Ershov & Samokhvalov (1984). Despite the generality of the above arguments, the mathematical result and revision of the foundations of mathematics obtained on the basis of this analysis in Ershov & Samokhvalov (1984) is its immediate consequence.

*"I'm thirsty – what does it mean? There is, of course, no mistake in thinking that the words "I am thirsty" simply mean this, where it is a certain state of consciousness that I am experiencing now and that I call thirst. But then a new question arises: how does the feeling of thirst (wishing) relate to actual drinking (satisfying wishing)? How do I know that drinking can satisfy your thirst? Does the experience of thirst itself contain a consciousness of how this thirst can be satisfied? ... **To know desire does not mean to know what is desired, but to know the ability to know what is desired** as soon as the opportunity presents itself. In other words, you understand a desire ... only when you have matched that desire with a sense of confidence that you will be able to recognize any future state of consciousness in a convincing and unmistakable way as a state of desire satisfaction or a state of dissatisfaction... Although ... I don't necessarily know how this quenching will be achieved. In my past experience, I expect it to be water, but maybe some pill will also quench my thirst".*

This argument allows us to clarify the ***tasks*** concept. "We understand a problem only when we compare it with a reasonable sense of confidence that we will be able to recognize any state of our consciousness in a convincing and unmistakable way as such when a solution is found, or as such when a solution is not found" Ershov & Samokhvalov (1984). Note that if the latter condition is not met, then the problem does not require a solution, since then any state of consciousness can be considered a solution.

Suppose we have some text. Is it a "convincing and error-free" presentation of the task solved? In mathematical theories, it is generally accepted that a "reasonable sense of confidence" that the statement of the solution of the task indeed its solution arises when this statement is a proof of the task. The proof provides a formal criterion for having a task solution. Let us assume that our states of consciousness, together with the evidence, can be formalized within the framework of some formal system S. Let us ask ourselves: does this formal system allow us to determine, by means of the formal system S itself, whether it is a proof of the solution of the problem or not? If such a formal system exists, it means that it can serve as a formal model for setting and solving mathematical problems. This question was analyzed in Ershov & Samokhvalov (1984; 2007) and it was proved that only in "weak" formal systems, in which Godel's incompleteness theorem does not pass, we can always determine by means of the formal system itself whether a certain text is a proof of the solution of a certain problem or not.

This result allowed its authors to formulate a new approach to the foundations of mathematics, consisting in a radical change in Hilbert's program of justification of mathematics. "As you know, Hilbert believed that, generally speaking, not all statements of any mathematical theory make sense. At the same time, he implicitly assumed that the division of the set of all statements of the theory under consideration into meaningful ("real") and meaningless ("ideal") is completely determined by the type of statements themselves and, therefore, is fixed for all theories with the same syntax and signature. According to the new paradigm, this division into meaningful and meaningless statements depends not only on the syntax and signature of the theory under consideration, but also on the class of problems that this theory is intended to deal with. From this point of view, the same theory as a mathematical calculus will have different sets of meaningful statements if it is designed to

handle different classes of problems. In other words, mathematical theory is considered simply as a "reservoir" for poorer formal systems, which are separately "extracted" from the whole theory depending on a particular problem at hand. By itself, regardless of possible tasks ... the theory has no practical significance, and therefore the question of whether it is contradictory in general or not is of no independent interest" Ershov & Samokhvalov (1984).

But we are not only interested in mathematical problems. We reformulate the concept of a task so as not to appeal to states of consciousness. We will say that *a task is meaningful* if and only if we have *a criterion for solving the task*, in the sense that for each proposed solution we are always able to determine whether it is a solution or not. Tasks in this sense arise not only in mathematics, but also in many other areas, and therefore in all these cases it should be borne in mind that a criterion for task solving is always necessary.

# 9   PURPOSE AND PURPOSEFUL ACTIVITY

Desire is active – it forces the body to show its activity in aimed at satisfying its desire. Then the concept of *goal* arises. You can't achieve a goal without having a criterion for achieving it, otherwise you can always can assume that the goal has already been achieved. The criterion for achieving the goal is the satisfaction of the desire. The concept of a goal is more general than the concept of a task – the goal of the task is its solution and the criterion for achieving this goal is the criterion of task solution.

Defining a goal doesn't make sense without a criterion for achieving it, because we need to make sure that the criterion is not met right now, and, therefore, it makes sense to set a *goal like what we don't have right now and what we want to achieve*. This definition of the goal allows us to define *the result* of achieving the goal, as all that we get when the criterion is met and the goal is achieved (desire satisfaction). There is the following relationship between the concepts of goal and result: the result is obtained when the goal is achieved and the achievement criterion is "triggered". But when a goal is set, we have a goal, but we don't have a result.

The definition of a goal is paradoxical, since the activity/activity to meet a certain criterion does not fundamentally imply knowledge about what and how to achieve the goal. You can set a goal without defining how, what and when to achieve it. This paradoxical nature of the goal concept is called *the goal paradox*. As we will see from functional systems theory, brain activity in goal-directed behavior is constantly focused on resolving the goal paradox and determining how, what and when the goal can be achieved.

The action is always purposeful. If there is no goal for the action, then it is not clear when (or how) it should end. The meaning of activity is to change the current state in order to achieve something. Purposeful activity is aimed at satisfying a certain need (desire) of the body.

# 10   FUNCTIONAL SYSTEMS THEORY

Functional Systems Theory is only one in which the concepts of goal, result and purposeful activity are central and where the physiological mechanisms of goal, result and purposeful activity are analyzed. Functional Systems Theory (TFS) is a theory of how the brain works as a system for achieving goals and resolving the goal paradox. Therefore, we will present the theory of functional systems as a theory of brain resolution of the goal paradox, which describes how the brain determines: how, what and when the goal can be achieved.

P.K. Anokhin wrote: "Perhapsone of the most dramatic moments in the history of studying the brain as an integrative education is the fixation of attention on the action itself, and not on its results ... we can assume that the result of the "grasping reflex" will not be grasping itself as an action, but that set of afferent stimuli that corresponds to the signs of the "grasped" object" Anokhin (1978; 1974; 1984). The "set of afferent stimuli" is the criterion for achieving the goal in the TFS. It should be noted that this dramatic moment persists to nowadays.

**Architecture of functional systems**. According to P.K. Anokhin, the central mechanisms of functional systems that provide purposeful behavioral acts have the same architecture.

*Afferent synthesis*. The initial stage of a behavioral act of any degree of complexity consists of afferent synthesis, which includes the synthesis of motivational arousal, memory, situational and trigger afferentation.

*Motivational arousal*. Goal setting is carried out by the need that has arisen, which is transformed into motivational arousal.

*Memory*. Motivational arousal "extracts from memory" all possible ways to achieve a goal, as well as the entire sequence and hierarchy of results that must be obtained in order to achieve the goal in some specific way.

**Situational afferentation**. When a trace is recorded in memory, the environment in which the result was obtained is also recorded. This environment is recorded as the necessary conditions, along with the motivation required to achieve results. Therefore, motivational arousal in a given situation "extracts from memory" only those ways of achieving the goal that are possible in this situation. Thus, situational afferentation, when interacting with the experience extracted from memory, determines what and how can be done in this environment to achieve the goal.

**Trigger afferentation**. In its meaning, it is also a situational afferentation, only it is not related to the stimuli of the situation, but to the time and place of achieving the goal. Therefore, trigger afferentation answers the question: when and where can the result be achieved?

Thus, at the stage of afferent synthesis, the goal paradox is largely resolved and it is determined what, how, where and when can be done to achieve the goal.

*Decision making*. At the stage of afferent synthesis by motivational arousal it can be extracted from memory (in a given situation) several ways to achieve a goal. At the decision-making stage, only one of these methods is selected – a *specific action plan*. By "pulling" all the accumulated experience from memory, motivational arousal is transformed into a *specific goal* that determines the way to achieve it. A specific goal is called "*higher motivation*" in the TFS.

*Acceptor of action results*. Motivational arousal also "extracts from memory" the entire sequence and hierarchy of results that must be achieved in order to complete the action plan. This sequence is called an *acceptor of action results* in the TFS. The acceptor of action results is the dominant need (Goal) of the body, transformed into the form of advanced brain stimulation, as if into a kind of *complex receptor* for future reinforcement, which is *a criterion for achieving a specific goal*.

*Reinforcements*. *The sanctioning stage*. If all the results of the acceptor of action results are achieved and the goal is achieved, then the last sanctioned stage occurs, in which the need is met and the specific action plan is stored in memory.

A formal model of TFS in terms of maximally specific causal connections (neurons) is given below.

## 11 CAUSALITY AND NEUROPHYSIOLOGY OF FUNCTIONAL SYSTEMS THEORY

Let us consider the TFS model that follows from neurophysiological data on neuronal excitations obtained in the TFS. Neurophysiologically, anticipation is realized by special collateral branches from the actions performed, which come to the "input" of the brain, converging with afferentation from input stimuli: "*We are talking about collateral branches of the pyramidal tract, which divert "copies" of those efferent messages that go to the pyramidal tract to many interstitial neurons ... Thus, the moment of decision-making and the beginning of the output of working efferent excitements (the beginning of actions – E.E.) from the brain is accompanied by the formation of an extensive complex of excitements consisting of afferent signs of a future result and a collateral "copy" of efferent excitements that went to the periphery along the pyramidal tract to the working organs*" Anokhin (1984).

In fact, this means the development of conditional (causal) connections between the implementation of actions (efferent excitements) and the subsequent perception of the results of actions represented by their afferent characteristics (see Fig. 3). When we perform actions, we immediately send a conditional signal along the collaterals that we are about to receive afferentation about the results of these actions. This leads to the development of conditional (causal) relationships between actions

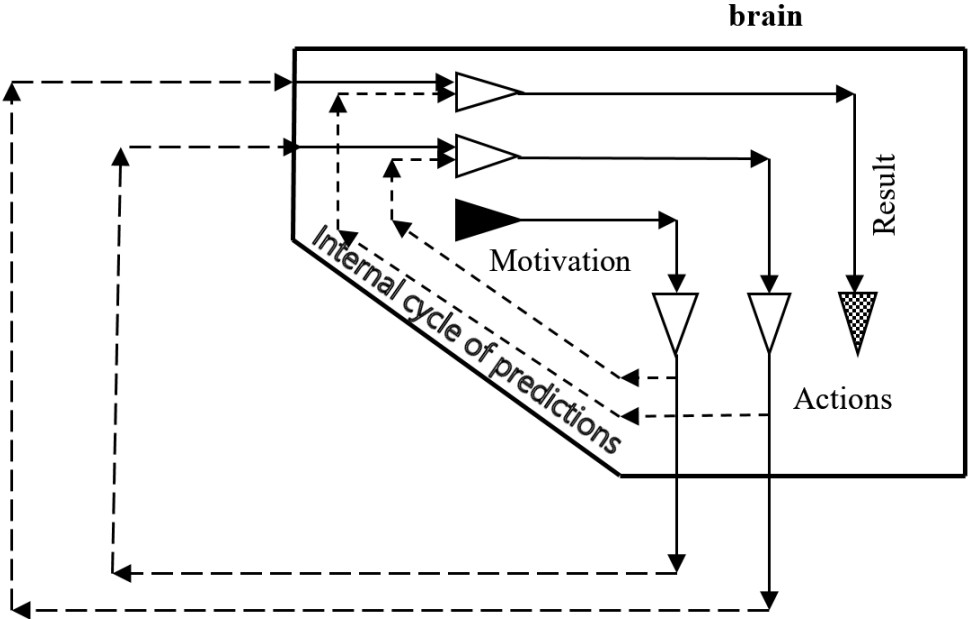

Figure 3: Formation of the inner contour of predictions

and their results, reflecting the relationship of actions and results occurring in the external world. These are conditional connections made by the brain along ***the internal circuit*** (see Fig. 3), allow you to predict the results of actions taking place in the external world, even before the results themselves appear. When motivational arousal activates various sequences of actions to achieve the goal, then at the same time, the entire sequence and hierarchy of results that will be obtained in the process of achieving the goal is predicted along the "inner contour". When a decision is made on a specific action plan, the achievement of all intermediate results that make up the acceptor of action results is simultaneously anticipated along the "inner contour".

## 12  TFS MODEL IN TERMS OF MAXIMALLY SPECIFIC CAUSAL RELATIONSHIPS

We expand the previous scheme in terms of maximally specific causal relationships and corresponding excitations Fig. 4.

We will consider the need as a request to the functional system to achieve the goal indicated by the predicate $PG_0$. This request is sent to the afferent synthesis block. For functional systems that do not have functional subsystems, it extracts causal relationships of the form

$$P_{i1}, \ldots, P_{im}, A_{k1}, \ldots, A_{kl} \Rightarrow PG_0$$

leading to goal achievement $PG_0$, where $P_{i1}, \ldots, P_{im}$ is the environment properties required to achieve the goal, and $A_{k1}, \ldots, A_{kl}$ is the sequence of actions leading to the goal. In this case the properties $P_{i1}, \ldots, P_{im}$ must be conformed with the properties of the environment $P_1, \ldots, P_n$ that enter the afferent synthesis block.

For hierarchically functional systems that call functional subsystems, this query extracts more complex causal relationships from memory

$$P_{i1}, \ldots, P_{im}, PG_{j1}, \ldots, PG_{jn}, A_{k1}, \ldots, A_{kl} \Rightarrow PG_0,$$

these include requests for achieving subgoals $PG_{j1}, \ldots, PG_{jn}$. Then the extracted rules are sent to the decision-making block, where the goal achievement forecast is performed for each of the rules. The forecast of the rules, where only actions are performed, is based on the probability of the rule itself. Forecasting based on rules with requests to subsystems is performed by sending these requests

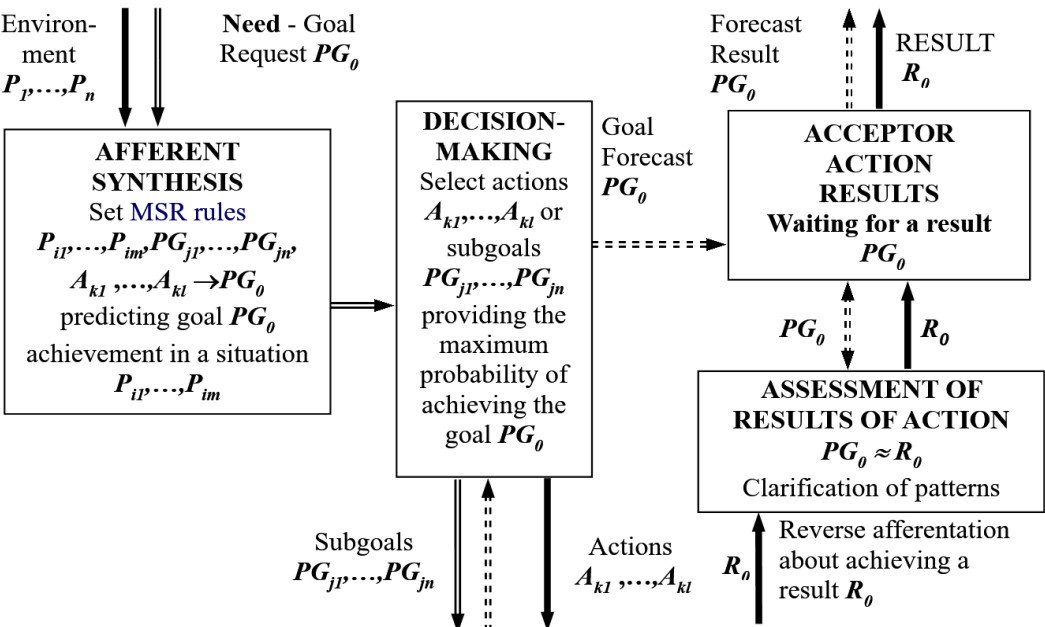

Figure 4: Scheme of the functional system

to these functional subsystems, making decisions in them and receiving probabilistic estimations of achieving these subgoals. The resulting forecast probability is calculated by multiplying the rule probability by the probability of achieving the subgoals.

After that, a decision is made for achieving the goal, and for this purpose, a rule is selected that has the maximum probability of predicting that the goal will be achieved. Then an action plan is formed that includes all actions included in the rule and all actions that are available in functional subsystems. Simultaneously with the action plan, the action results acceptor is formed, which includes the expectation of all predicted sub-results in functional subsystems and in the functional system itself. After that, the action plan is executed, and the expected results are compared with the results obtained.

If all the sub-results and the final result are achieved and coincide with the expected results, then the rule itself and all the rules of functional subsystems that were selected during the decision-making process are reinforced and their probability increases.

If the result is not achieved in a particular subsystem, the corresponding rule of this functional subsystem is penalized. After that, there is a tentative research reaction, which revises the decision to achieve the goal.

This model has been repeatedly refined and used for modeling animates Mukhortov et al. (2012); Demin & Vityaev (2014); Vityaev et al. (2020). The most revealing experiment was conducted with a nematode Demin & Vityaev (2014), when this model was embedded in an electronic model of the nematode and after learning by "trial and error", this model learned its natural movement, which is used by a real nematode.

Thus, the cognitive group of functional systems has been successfully formalized.

## 13    Conclusion

Thus, we were able to show that by taking as a basis the principle that *the brain detects all possible causal relationships in the external world and draws conclusions from them*, we can obtain mathematical models of the neuron, cellular ensembles in the form of probabilistic formal concepts and model of functional system.

These mathematical models are based on an accurate mathematical analysis of probabilistic causality and its formalization in the form of probabilistic maximally specific causal relationships, as well as on probabilistic generalization of formal concepts, which is a looping conclusion on maximally specific causal relationships.

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
