# OpenReview forum: "Mathematics of natural Intelligence"
_mathai.club/MathAI/2025/Conference — MathAI 2025 Oral_

### Official Review · Reviewer_PYba · 2025-02-25
**Mathematics of natural Intelligence**

**Rating:** 9
**Confidence:** 3

**Review:**

The paper proposes a probabilistic model of cause-and-effect relationships of cognitive processes reflecting the evolution of the brain. The basis of the research is the use of the theory of the Cognitom as a subject of mental activity for modeling.
This work presents models of: ”natural” classification; theory of functional brain systems by P.K. Anokhin; prototypical theory of categorization by E. Roche; theory of causal models by Bob Rehter; theory of consciousness as integrated information by G. Tononi.
 A detailed review of the listed theories is given, a mathematical model of the Cognitom and a formal model of the neuron are proposed.
The authors claim that a semantic interface and a software package implementing a neural model were also created, which were successfully used to solve a number of applied problems.
However, it should be noted that there is no description of the «Discovery» software package. There are also several inaccuracies in the presentation, for example, figures 5 and 6 (section 11) are missing, which the authors refer to in the presentation, which makes it somewhat difficult to read this section.
Overall, the work is certainly of interest as an attempt at a mathematical description of intellectual activity and contributes to the understanding of the possibilities of using neural networks as a means of modeling in artificial intelligence.

---

### Official Review · Reviewer_PB4B · 2025-02-25
**MATHEMATICS OF NATURAL INTELLIGENCE**

**Rating:** 7
**Confidence:** 3

**Review:**

In the paper, the authors showed that by taking as a basis the principle that the brain detects all possible causal relationships in the external world and draws conclusions from them, it can obtain mathematical models of the neuron, and cellular ensembles in the form of probabilistic formal concepts and model of functional system.  Based on this, the paper presents models of: ”natural” classification; the theory of functional brain systems by P.K. Anokhin; the prototypical theory of categorization by E. Roche; the theory of causal models by Bob Rehter; the theory of consciousness as integrated information by G. Tononi. There are many theoretical results and definitions. The paper provides many lemmas and theorems with proofs.

However, it should be noted that according to the conference rules, the number of pages allowed is up to 10. The article now with references contains 17 pages. The article can be arranged so that it includes the main results and the rest of the results and evidence can be placed in an appendix to the article.

Figures 5 and 6 in section 11 are missing.
In the first reference, the authors are missing.


There are many typos in the paper and in the following some of them
1. In the abstract, line 3. cognitome reads cognition
2. line 015. cognitome  -------> cognition
3. line 036. brain neural hypernetwork -------> brain's neural hypernetwork
4. line 057. statistically laws -------> statistical laws
5. line 083. as a fixed points -------> as fixed points
5. line 086. as a fixed points -------> as fixed points
6. line 099. On the highly  -------> The highly
7. line 099.  is also base the -------> is also based on the
8. line 104.  it cannot say that concept -------> it cannot be said that the concept

---

### Official Review · Reviewer_QixZ · 2025-02-27
**The work solves an important problem of modeling natural intelligence and can be recommended for publication.**

**Rating:** 9
**Confidence:** 4

**Review:**

The paper presents a mathematical framework for modeling natural intelligence based on the concept of cognitome introduced by K. V. Anokhin. Cognitome is described as a high-order brain structure consisting of interconnected cognitive groups of neurons (COGs), which include functional systems and cellular ensembles. The authors suggest that the brain functions by discovering all possible cause-and-effect relationships in the external world and drawing conclusions from them. This principle is used to derive mathematical models of various cognitive processes. The paper introduces several new elements, including probabilistic formal concepts, maximally specific cause-and-effect relationships, and a formal model of functional systems, which distinguish it from previous studies. The authors' rethinking of integrated information theory and their sequential neuron model further contribute to the development of cognitive science and neuroscience. Overall, the work is of great practical and theoretical importance and can be recommended for publication.

---

### Decision · Program_Chairs · 2025-03-08

**Decision:**

Accept (Oral)

**Comment:**

Your article has been accepted and you can make a presentation on the article. All articles will be sorted by rating and within the available conference places one author from each article will be invited. If there are not enough places, then you will either have the opportunity to present remotely or come at your own expense!